# Reconfigurable soft body trajectories using unidirectionally stretchable composite laminae

Sang Yup Kim [1,4], Robert Baines [1,4], Joran Booth[1], Nikolaos Vasios[2,3], Katia Bertoldi[2,3] & Rebecca Kramer-Bottiglio[1]

Compliant, continuum structures allow living creatures to perform complex tasks inaccessible to artificial rigid systems. Although advancements in hyper-elastic materials have spurred the development of synthetic soft structures (i.e., artificial muscles), these structures have yet to match the precise control and diversity of motions witnessed in living creatures. Cephalopods tentacles, for example, can undergo multiple trajectories using muscular hydrostat, a structure consisting of aggregated laminae of unidirectional muscle fibers. Here, we present a self-adhesive composite lamina inspired by the structural morphology of the muscular hydrostat, which adheres to any volumetrically expanding soft body to govern its motion trajectory. The composite lamina is stretchable only in one direction due to inextensible continuous fibers unidirectionally embedded within its hyper-elastic matrix. We showcase reconfiguration of inflation trajectories of two- and three-dimensional soft bodies by simply adhering laminae to their surfaces.

[1] School of Engineering and Applied Science, Yale University, New Haven, CT 06511, USA. [2] Harvard John A. Paulson School of Engineering and Applied Sciences, Harvard University, Cambridge, MA 02138, USA. [3] Kalvi Institute of Bionano Science & Technology, Harvard University, Cambridge, MA 02138, USA. [4] These authors contributed equally: Sang Yup Kim, Robert Baines. Correspondence and requests for materials should be addressed to R.K.-B. (email: rebecca.kramer@yale.edu)

The introduction and critical study of hyper-elastic materials have sparked an evolution in numerous domains of science and engineering, such as medicine[1,2], electronics[3–5], and robotics[6–8]. The inherent stretchability and resilience of hyper-elastic materials make them apt for incorporation into devices imitating the physiology of living creatures. In robotics, hyper-elastic materials allow for the creation of compliant, continuum soft robots with greater adaptability and flexibility than conventional linkage-joint robots[9–13]. Many soft robots inspired by the morphology of biological organisms have performed movements of unprecedented complexity. A typical mechanism to impart such sophisticated movement is the incorporation of strain limiters onto a soft body. When the body inflates, strain limiters exert contraction forces analogous to biological muscle fibers, governing specific shape changes[14–20]. McKibben and PneuNet actuators are pneumatic artificial muscles widely used in practice, and exploit confinement in woven mesh[21,22] or by rigid compartments, as strain limiters[23–25]. However, these actuators are often suboptimal for use in entirely-soft robots: McKibben actuators are difficult to integrate into soft bodies without rigid fixtures, and PneuNets are challenging to predict and control due to their intricate geometry. Recent studies have established a simple yet effective way to control the deformation trajectory of soft actuators—by wrapping them with continuous inextensible fibers[26–29]. Varying the angle of wrapped fibers creates unique strain-limiting patterns that change how an internal elastomeric bladder expands, and allows for predictable extension, twisting, and bending. However, in this approach, programmed directionality is permanent and constrained to one fixed motion, unlike the versatility exhibited by biological organisms. Additionally, fiber-wrapped actuators have tedious manufacturing processes that make them difficult to integrate with two-dimensional (2D) geometries or scale up.

Looking to nature for inspiration in addressing the aforementioned challenges with existing soft actuators, we noticed that remarkably dexterous motion achieved by skeleton-free animal parts such as elephant trunks, octopus arms, and human tongues, is attributed to muscular hydrostats: a laminated structure composed of layers of unidirectional muscle fibers[30]. Selective contraction of individual muscle layers in three primary axes—transverse, longitudinal, and oblique—grants muscular hydrostats the ability to move precisely and with infinite degrees of freedom[31,32]. For example, cephalopod tentacles twist by contracting oblique muscle fibers and bend by simultaneously contracting transverse and longitudinal muscle fibers (Fig. 1a). A simple superposition of these unidirectional muscle contraction primitives gives rise to sophisticated deformation.

Inspired by the fiber architecture of the muscular hydrostat, we developed a lamina composed of a hyper-elastic matrix with unidirectionally embedded inextensible fibers and an adhesive backing (Fig. 1b). We call this composite material Stretchable Adhesive Uni-Directional prepreg (STAUD-prepreg). STAUD-prepreg is extremely stretchable in the direction perpendicular to the fibers and inextensible along the fibers, boasting a 1000-fold difference in stiffness and elongation (Fig. 1c). When STAUD-prepreg is adhered to volumetrically expanding soft bodies, the embedded inextensible fibers induce selective contraction forces analogous to muscle fibers in the muscular hydrostat, and govern shape change (Fig. 1d, Supplementary Movie 1). The fabrication of STAUD-prepreg is simple and scalable, leveraging a fiber-winding process that is already established in industry for mass-production of industrial-grade composites to create large 2D sheets, which can be cut and applied to almost any soft actuator. Namely, we use a bench-top fiber winder wherein a mixture of two-part silicone resin is infused into polyester fibers uniformly spaced on a rotating mandrel (Fig. 1e). We then apply a silicone-based adhesive to the back of the lamina after the resin fully cures to create a self-adhesive layer. The deformation trajectories of soft bodies are easy to program by placing STAUD-prepreg on their surfaces (Fig. 1f). Adhered STAUD-prepreg can be easily detached and re-attached in various patterns and stacking sequences to reconfigure the inflation trajectories.

## Results

**Influence of STAUD-prepreg on soft actuator trajectories.** To reveal the influence of STAUD-prepreg on morphing trajectories of soft bodies, we began by constructing a freely-inflating cylindrical actuator wrapped in one STAUD-prepreg. By adjusting the fiber orientation of this STAUD-prepreg, the cylindrical actuator exhibits a variety of motions such as contraction, elongation, and rotation (Fig. 2a). Contraction (Fig. 2a at 0°, Supplementary Movie 2) is accomplished when the STAUD-prepreg is wrapped at $\theta = 0°$; that is, embedded fibers are aligned longitudinally to the cylinder length. The 0° fibers geometrically constrain the pneumatic cylinder such that it is only capable of expanding transversely, resulting in length-wise contraction. Elongation (Fig. 2a at 90°, Supplementary Movie 3) is achieved with the STAUD-prepreg fibers oriented at $\theta = 90°$; that is, embedded fibers are aligned perpendicular to the cylinder's length. The 90° fibers prevent the actuator from expanding transversely, forcing it to extend axially. Rotation (Fig. 2a at 45°, Supplementary Movie 4) occurs in a manner similar to elongation, except the angled fibers ($\theta = 45°$ in this study) exert a torque with respect to the central axis. In all cases, the deformation of a cylindrical actuator is proportional to the input pressure and directly affected by the fiber spacing of the adhered prepreg. Moreover, STAUD-prepreg are capable of programming inflation trajectories of 2D objects and controlling their shape in 3D space. For instance, applying STAUD-prepreg to a thin, planar soft body elicits bending upon its inflation (Fig. 2b, Supplementary Movie 5). Planar actuators outfitted with STAUD-prepreg are light and occupy a small footprint when not actuated, making them ideal for deployable, easily-transportable pneumatic devices. In general, the initial shape of soft bodies (i.e., 2D vs 3D cylinder) influences more the adherence of the STUAD-prepreg, than it does the uniqueness of a final shape. The 2D configuration of STAUD-prepreg is easier to adhere to 2D bodies than 3D ones without void entrapments that can potentially degrade the adhesion quality.

**Mechanical characterization of STAUD-prepreg.** We conducted tensile mechanical testing of STAUD-prepreg laminae with different fiber orientations and spacings to ascertain its fundamental mechanical properties. Quasi-static, unidirectional tensile tests reveal a stark difference in stretchability (i.e., stiffness and elongation) of the material, depending on the loading direction (Fig. 2c). For example, the tensile modulus of the lamina (at 2 mm fiber spacing) is 56 MPa in the longitudinal direction of the fibers ($E_1$), and 0.065 MPa in the transverse direction ($E_2$). Moduli tend to increase with decreasing fiber spacing as a result of the higher fiber content in the lamina[33]. Theoretically calculated values from micromechanics modeling[34] confirm our experimental results for moduli:

$$E_1 = V_f E_f + V_m E_m \tag{1}$$

$$E_2 = (V_f/E_f + V_m/E_m)^{-1} \tag{2}$$

where $E_f$ and $E_m$ are tensile moduli of fibers and matrix, and $V_f$ and $V_m$ are the volume fraction of fibers and matrix, respectively (see "Supplementary method" for details).

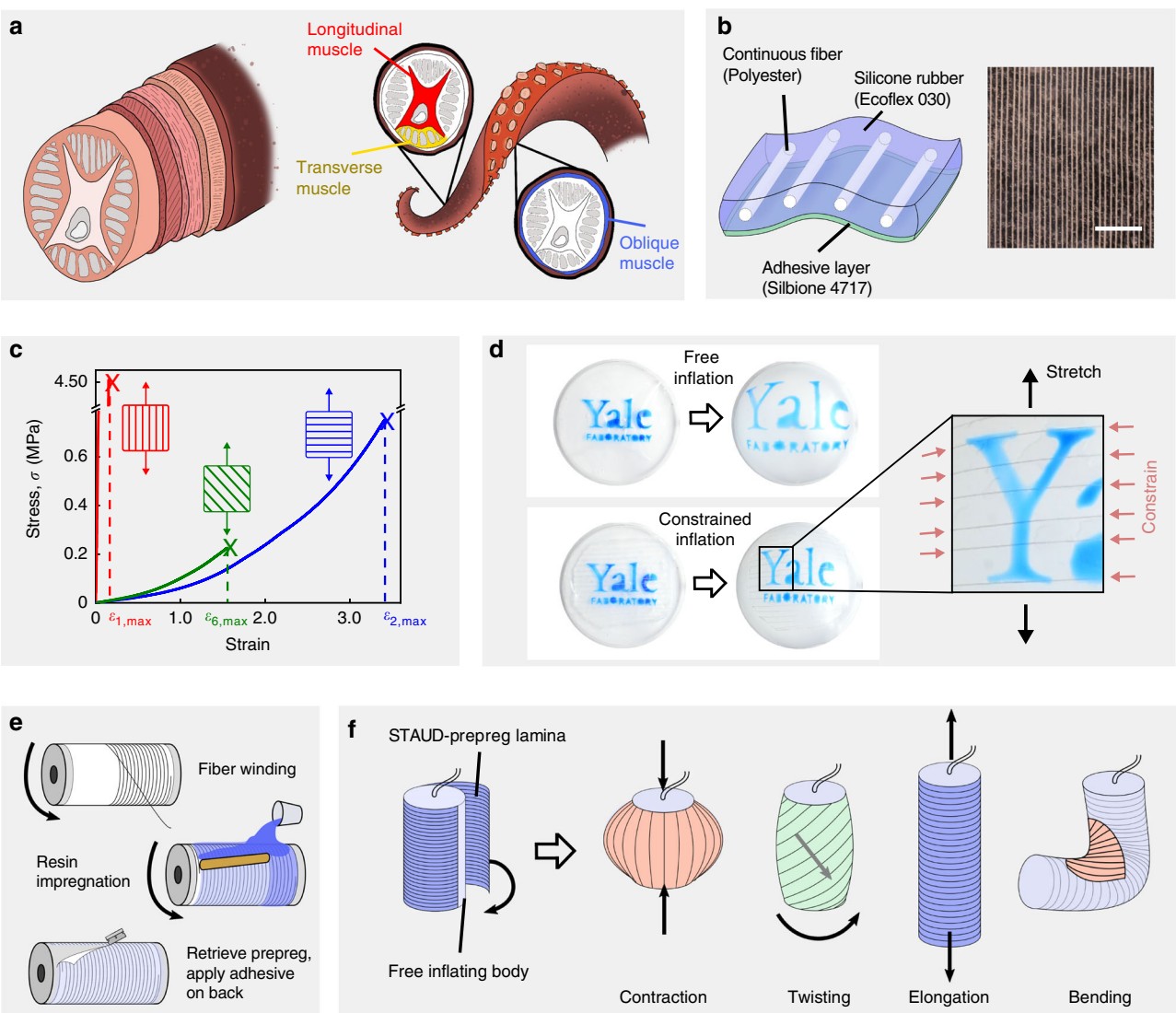

**Fig. 1** Overview and working principle of STAUD-prepreg. **a** Illustration of laminated layers of unidirectional muscle fibers in a muscular hydrostat in cephalopod tentacles and the working mechanism for achieving complex motion. **b** Schematic and optical micrograph of stretchable adhesive unidirectional prepreg (STAUD-prepreg). Scale bar: 10 mm. **c** Quasi-static uniaxial tensile testing results of STAUD-prepreg laminae with respect to different fiber orientations ($\theta = 0°$, 45°, 90° from left). **d** Photographs of inflating planar soft body. Free inflation (top) with uniform radial expansion and constrained inflation (bottom) with one adhered lamina at $\theta = 90°$. **e** Schematic of fabrication process for STAUD-prepreg using a bench-top fiber winder. **f** Illustration of cylindrical actuators with one STAUD-prepreg lamina adhered, and the ensuing inflation trajectory

**Analytic prediction of deformation.** To predict deformation of an inflating body outfitted with STAUD-prepreg, we used our experimental tensile test results to construct a stiffness matrix ($\mathbf{C}_{ij}$) of the lamina, for a relation $\sigma_i = \mathbf{C}_{ij}\varepsilon_{ij}$ ($i, j = 1, 2, 3, \ldots 6$), where $\sigma$ is stress and $\varepsilon$ is strain. Since STAUD-prepreg is thin and its properties are symmetric about one axis, the stiffness matrix is further simplified to $[\mathbf{Q}]_{x,y}$, for a relation $[\sigma]_{x,y} = [\mathbf{Q}]_{x,y}[\varepsilon]_{x,y}$, where $[\sigma]_{x,y}$ and $[\varepsilon]_{x,y}$ are stress and strain matrices in the x-y plane, respectively[35]. This stiffness matrix allows us to analytically predict morphing of the host body clad in STAUD-prepreg laminae, after a slight modification to classical laminate theory[36–38]:

$$\sum_{k=1}^{n} \int_{z_{k-1}}^{z_k} [\sigma]_{x,y}^{k}\,\mathrm{d}z = \sum_{k=1}^{n} \int_{z_{k-1}}^{z_k} [\mathbf{Q}]_{x,y}^{k}[\varepsilon]_{x,y}^{k}\,\mathrm{d}z \quad (3)$$

where $k$ is an individual lamina, $n$ is the total number of laminae, and $z$ is the out-of-plane coordinate (see "Supplementary

method" for details). A prediction model from the augmented classical laminate theory (ACLT) shows good agreement with the experimental strain values of the cylindrical actuators in Fig. 2a, using Eq. (3) at $n = 1$. Adapting ACLT to the soft robotics space facilitates the prediction of the morphing of soft bodies clad in STAUD-prepreg, and enables intuitive programming of specific motions.

**Influence of multiple STAUD-prepreg on soft actuator trajectories.** To impart more complex motion on a soft body, we created actuators with STAUD-prepreg patterns emulating the muscular hydrostat structure. For instance, we adhered multiple STAUD-prepreg in a stack, or cut a single STAUD-prepreg into various-sized patches, and distributed them across a soft body's surface. We used a cylindrical actuator in Fig. 2a at 90° as a base that forced elongation, and adhered additional STAUD-prepreg to further direct motion. When a STAUD-prepreg with $\theta = 0°$ was adhered in a way that covered the entire base, the actuator

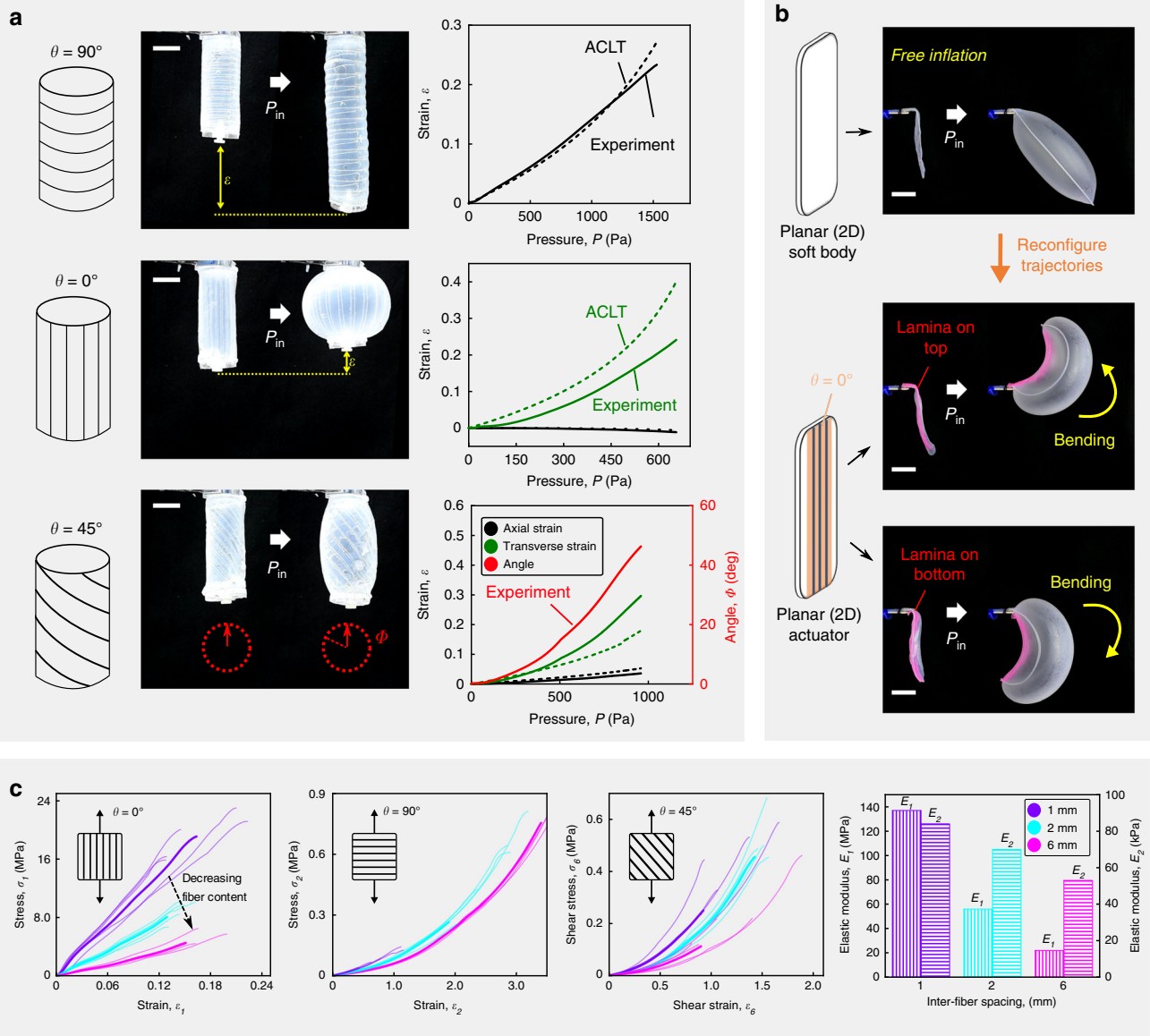

**Fig. 2** Basic programmed inflation trajectory of 2D/3D soft bodies and mechanical characterization of STAUD-prepreg. **a** Inflating cylindrical actuator wrapped with one STAUD-prepreg lamina at different fiber orientations of $\theta = 90°$, $0°$ and $45°$. Experimental results are juxtaposed with analytical results from augmented classical laminate theory (ACLT). Legend color codes apply to all plots. Scale bar: 25 mm. **b** Planar soft body consisting of two silicone rubber films and reconfiguration of its inflation trajectory by adhering one STAUD-prepreg. Scale bar: 25 mm. **c** Quasi-static uniaxial tensile testing results for STAUD-prepreg. $E_1$ and $E_2$ denote the Young's modulus of the STAUD-prepreg along the fibers ($\theta = 0°$) and perpendicular to the fibers ($\theta = 90°$), respectively. Higher fiber content in the prepreg results in lager values of $E_1$ and $E_2$. $E_1$ is ~1000-times greater than $E_2$

underwent virtually no motion due to the presence of strain-limiting fibers preventing extension in both principal directions (Fig. 3a). Replacing the adhered STAUD-prepreg ($\theta = 0°$) with one at $\theta = 45°$ gave rise to rotational motion with a negligible expansion in volume, unlike the actuator with one lamina adhered at $\theta = 45°$ (see Fig. 2a at 45°). The ACLT model at $n = 2$ in Eq. (3) shows good agreement with experimental results and confirms the utility of ACLT in predicting the deformation of actuators clad in STAUD-prepreg. When a bulk STAUD-prepreg is cut into patches and adhered to the elongating base actuator, we coerce deformations that cannot be attained with actuators whose surfaces are entirely covered by STAUD-prepreg (see Figs. 2a, 3a), and unlock a larger workspace without any significant jumps in mechanical complexity. For example, bending, the underlying motion for grasping and locomotion in cephalopods, is achieved in our system when one STAUD-prepreg patch

is adhered to locally confine elongation (Fig. 3b, Supplementary Movie 6). Adjusting the fiber orientation of the patch results in a compound bending and twisting motion (Fig. 3c). Further complex output motion arises when multiple STAUD-prepreg patches are attached to the actuator in different locations. Each adhered patch creates contraction forces on the actuator based on its respective dimensions (i.e., $w$ and $h$) and elicits unique trajectories (Fig. 3d).

**Numerical modeling of deformation.** We performed finite element (FE) analysis on cylindrical actuators' motion using ABAQUS commercial software (see "Supplementary method" for details). FE simulation results confirm our experimental results and reveal detailed strain and stress fields on the inflating actuator (Fig. 3e as an example for the bending motion in

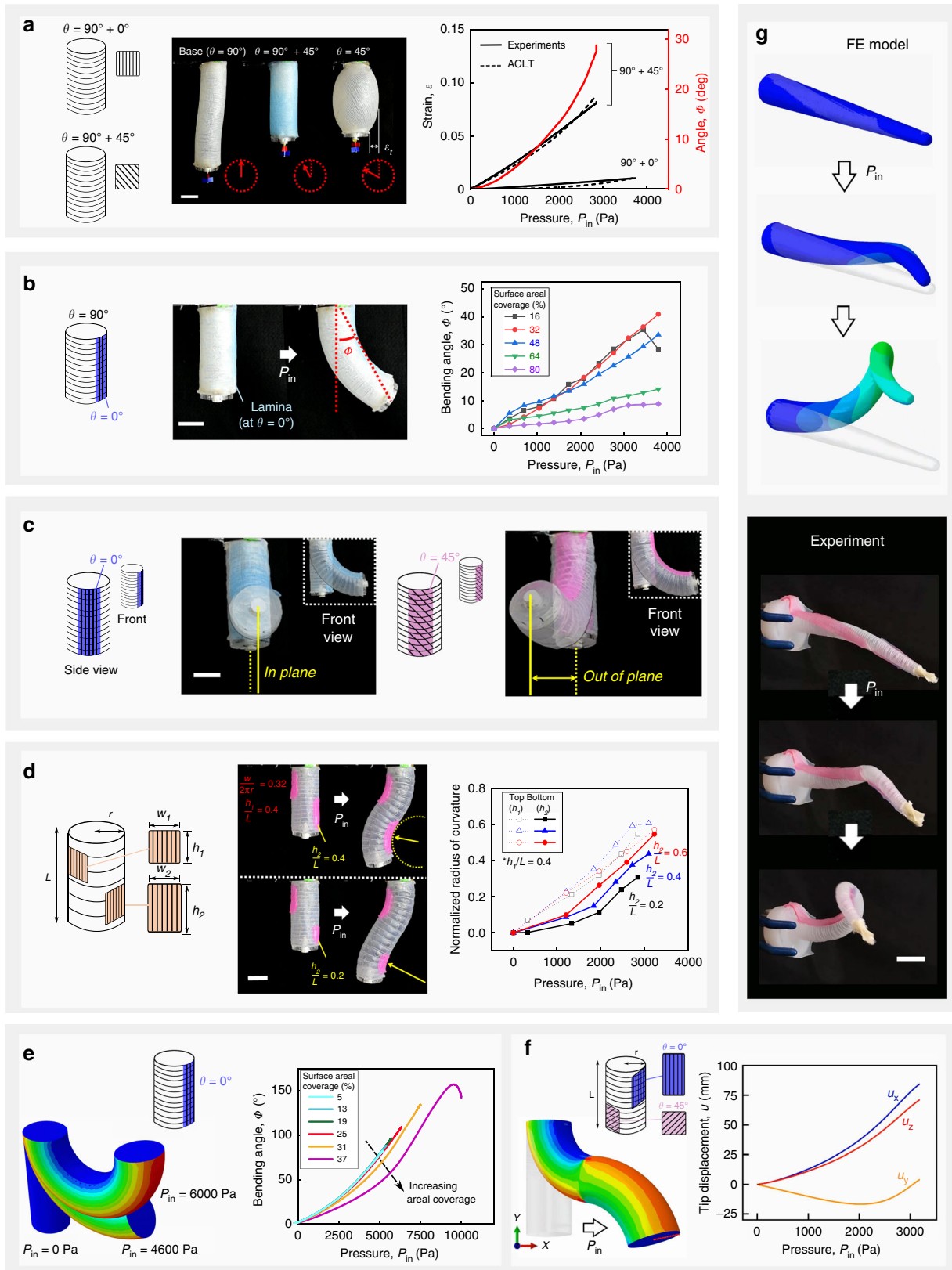

Fig. 3b). Moreover, FE modeling allows prediction of the influence of localized patch(es) on a soft body otherwise too complicated for ACLT to predict (Fig. 3f). We utilized the FE modeling to obtain STAUD-prepreg patch designs that when adhered to a slender soft body elicit dexterous movements similar to cephalopods tentacles. We then reproduced the simulated results in the laboratory (Fig. 3g, Supplementary Movie 7). Experimental results show good agreement with the simulation, testifying to the fact that FE modeling serves as an accurate design tool.

**Fig. 3** Complex programmed inflation trajectory of 3D soft bodies and finite element simulation. **a** Elongating cylindrical actuator (at $\theta = 90°$) with an additional STAUD-prepreg wrapped around at $\theta = 0°$ or 45°. The addition of a STAUD-prepreg at $\theta = 0°$ immobilizes the actuator, while one at $\theta = 45°$ gives rise to rotational motion without transverse strain ($\varepsilon_t$). Scale bar: 25 mm. **b–d** Bending motion of a cylindrical actuator due to a segmented STAUD-prepreg patch. **b** Increasing the areal coverage of the patch decreases the bending angle at a given pressure. **c** Adhering a patch at $\theta = 45°$ elicits a mixed motion of bending and twisting. **d** Two-localized patches with increasing patch dimension along the actuator length leads to a higher bending curvature. Scale bar: 25 mm. **e, f** Finite element (FE) analysis results using ABAQUS on cylindrical actuators. **e** Bending angle as a function of input pressure and a patch's areal coverage, as gathered from the simulation. **f** Two-localized patches with different fiber angles (at $\theta = 0°$ and 45°). **g** A slender cylindrical actuator mimicking the motion of cephalopod tentacles. FE simulation of the actuator with particular patch pattern and dimensions matches with the experiment. Scale bar: 50 mm

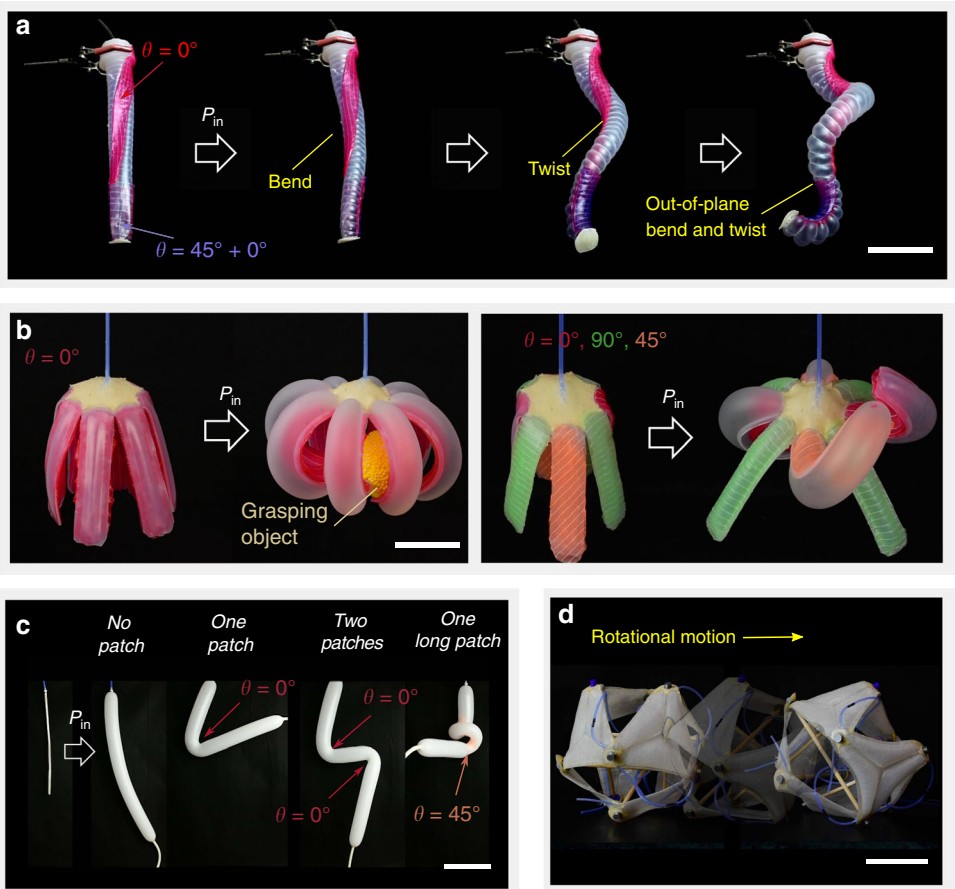

**Fig. 4** Practical use cases for STAUD-prepreg controlling compliant systems. **a** Life-like complex grasping motion akin to cephalopod tentacles accomplished by a cylindrical actuator. Scale bar: 100 mm. **b** Planar soft body with eight tentacles for use as a pop-up gripper. Adhering STAUD-prepreg patches gives rise to specific movements of individual tentacles that previously inflated without directional bias. Scale bar: 50 mm. **c** Conventional balloon with adhered STAUD-prepreg patches. The fiber orientation and dimension of the applied patches determine the inflation trajectories of the balloon. Scale bar: 50 mm. **d** A tensegrity robot consisting of rigid struts and soft membrane actuators made from STAUD-prepreg. A combination of rigid components and soft, planar actuators allows this hybrid system to roll from face to face by inflating the membrane actuators to locally destabilize itself. Scale bar: 200 mm

**Demonstrations using STAUD-prepreg.** We demonstrated several practical use cases for STAUD-prepreg. First, linearly scaling up the dimensions of the actuator in Fig. 3d gives rise to a life-like grasping motion; a combined movement of wrapping and bending reminiscent of cephalopod tentacles (Fig. 4a, Supplementary Movie 7). We then created a planar actuator consisting of eight tentacles that inflate uniformly without directional bias. Attaching STAUD-prepreg patches of varying fiber orientation enabled us to individually control the tentacles' shapes during inflation (Fig. 4b, Supplementary Movie 8). The planar actuator outfitted with STAUD-prepreg is capable of grasping and twisting an object, although it is significantly lighter and more compactable compared to conventional soft grippers. The versatility of STAUD-prepreg is further demonstrated by its application to every-day soft bodies, such as balloons (Fig. 4c, Supplementary Movie 9). When adhered to a balloon, STAUD-prepreg patches govern its inflation trajectory based on the quantity, dimensions, and fiber orientations of the patches. This reaffirms how balloons—cheap and ubiquitous as they are—could be used as programmable actuators in soft robots. Moreover, we made planar membrane actuators to change the shape of skeletal robotic systems. In particular, we created thin elastomer membranes outfitted with a layer of STAUD-prepreg to interface the compressive members of a tensegrity robot. The membrane

actuators are positioned on the exterior faces of the tensegrity, providing ample internal space for the robot to deform and roll when the actuating-faces inflate (Fig. 4d, Supplementary Movie 10).

We have shown that STAUD-prepreg is a versatile material capable of transforming entirely-soft bodies and hybrid soft-rigid architectures into customized, programmable, and re-configurable deforming systems. The effect of adhered patches on inflating soft bodies is easy to predict due to the simple strain-limiting behavior of the embedded unidirectional fibers. We verified that the shape change of actuators outfitted with STAUD-prepreg is analytically predictable using ACLT, and utilized numerical modeling in ABAQUS to predict compounded motion. Ultimately, we demonstrated highly complex, predictable shape change of 2D and 3D soft bodies reminiscent of the fiber-directed motion of a muscular hydrostat. The fundamental mechanisms of localized, reconfigurable strain-limiting fibers presented herein take a step closer to devising soft robot actuation ever more like real-life fiber musculature that can modulate directional contractions in situ. We expect STAUD-prepreg to become an essential component for controlling synthetic all-soft bodies, but also foresee its adoption into sports, medicine, and rehabilitation spaces due to its ease of implementation and manufacture.

## Methods

**Materials**. To create an elastomer matrix for the lamina, elastomeric platinum-cure resin (Ecoflex 00–30, Smooth-On) was mixed in a 1:1 ratio by weight. The mixture was subjected to 2000 rpm for 1 min in a planetary mixer (Thinky ARE-310) and degassed at 2200 rpm for an additional minute. The pneumatic cylinders, planar actuators, and membrane actuators used in the demonstrations were also fabricated from this resin, to create soft base bladders sufficiently compatible for the STUAD-prepreg to remain adhered. Polyester continuous fibers (100% Spun Thread, PRC) were used as received. The resin for the self-adhesive backing of the lamina was created by mixing a 1:1 weight ratio of silicone adhesive constituent (Silbione 4717, Elkem Silicones). Tabs for uniaxial tension tests were made from muslin fabric (Product #: 8808K11, McMaster Carr) infused with a platinum-cure resin (Dragon Skin 10, SmoothOn).

**Prepreg fabrication**. We built a fiber-winding machine to fabricate STAUD-prepreg with tunable properties in a scalable fashion. The fiber winder consists of a spool holster, a linearly-translating fiber distributor, a rotating winding mandrel, and controlled via an Arduino with an attached Adafruit V2 motor shield. To create bulk lamina, we entered the desired fiber spacing into the Arduino code. We then secured the polyester fiber spool into its holster adjacent to the fiber-winding machine. Next, the machine automatically wound polyester fibers onto the rotating drum. Subsequently, the elastomer resin was poured onto the fibers and smoothed with a resin scraper to disperse it evenly. We allowed the drum to keep rotating for 12 h to allow the resin to disperse evenly as it cured. A self-adhesive layer was then created on the back of the fully-cured prepreg by applying silicone adhesive that cures at room condition for 12 h. This self-adhesive layer was previously reported to maintain its adhesive force up to 20 attachment/detachment cycles, depending on the preparation of target surfaces[39]. We note that sufficient areal coverage between the adhesive layer and an inflatable body is required for robust attachment. For example, in Fig. 3b the adhesion of the STUAD-prepreg to a cylindrical actuator to induce bending motion was reliable with areal coverage >16%.

**Augmented classical laminate theory**. We constructed a simplified compliance matrix in the fashion that considers both geometric non-linearity and material non-linearity of fiber-reinforced elastomers (Supplementary Eq. 12). Furthermore, in line with the original classical laminate theory, we account for the angle of fibers of each lamina in a laminate by subjecting them to a transformation matrix (Supplementary Eq. 13). Lastly, we consider the effect of cyclic strain relaxation of STAUD-prepreg via a ratio of regression equations gathered from cycled and un-cycled STAUD-prepreg stress-strain curves. The final formulation of ACLT is written as:

$$\sum_{k=1}^{n}\int_{z_{k-1}}^{z_k}[\mathbf{e}]_{x,y}^k \mathrm{d}z = \Gamma_c \left[\sum_{k=1}^{n}[\mathbf{S}]_{x,y}^k{}^{-1}\right]^{-1}\sum_{k=1}^{n}\int_{z_{k-1}}^{z_k}[\boldsymbol{\sigma}]_{x,y}^k \mathrm{d}z \qquad (4)$$

Here, $\mathbf{e}$ and $\boldsymbol{\sigma}$ are Eulerian terms referring to the current principal material coordinates, $\Gamma_c$ is a cycle-depending scaling factor to adjust for strain relaxation, $\mathbf{S}$ is a simplified compliance matrix which relates in-plane stresses to in-plane strains for laminae, all for an $n$ layered-laminate, where each layer $k$ has thickness

$z_k - z_{k-1}$ and its mid-plane is a distance $z_k$. The input to the model is a stress term obtained by converting pressure to stress assuming the cylindrical actuators can be modeled as expanding pressure vessels. This model is also supplemented by geo-metrical boundary conditions outlined in the supplemental document. Results in Figs. 2a and 3a show good agreement between experimentally observed and ana-lytically predicted deformation of the various lamina-wrapped cylindrical actuators.

**Finite element analysis**. We performed finite element analysis (FEA) simulations using Abaqus (SIMULIA, Providence RI) commercial software in order to fully resolve the strain and stress distributions of each laminae and further account for boundary effects in the various cylindrically shaped actuators. FEA simulations allowed us to predict the deformation of more complex geometrical layouts and localized patches of STAUD-prepreg, and to subsequently replicate these simulated movements and designs in real-life systems (Fig. 3d; Supplementary Movie 7). All simulated models were generated using shell elements using the composite shell section formulation that enables numerical integration of composite shells con-sisting of several laminae in various orientations. Each STAUD-prepreg lamina was modeled as an anisotropic continuum 3D shell whose constitutive behavior is governed by the Holzapfel-Gasser-Ogden (HGO) anisotropic hyper-elastic continuum model.

## Data availability

The data in support of the findings and detailing the studies are available from the corresponding author upon request.

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

## Acknowledgements

We thank Dr. Michelle C. Yuen for assisting with demonstrations. This work was supported by a Small Business Technology Transfer grant (No. 80NSSC17C0030) from the National Aeronautics and Space Administration and an Office of Naval Research Young Investigator award (No. N00014–17–1–2604).

## Author contributions

S.Y.K. and R.K.B. conceived the idea. S.Y.K. and R.B. performed the manufacturing. R.B. and N.V. performed the material characterization and modeling. S.Y.K., R.B. and J.B. performed the demonstrations. K.B. and R.K.B. reviewed and commented on the manuscript. All authors participated in writing the manuscript.

## Additional information

**Competing interests:** The authors declare no competing interests.

