## [Peer Review File · Nature Communications]

Reviewers' comments:

Reviewer #1 (Remarks to the Author):

The paper presents an interesting laminated structure that can be applied to pneumatically-actuated soft robots in order to guide their deformation, thus the resulting movement of their actuation. The principle of using fibers to constrain and guide the deformation of pneumatically-actuated soft robots is not new per se, but the idea of building adhesive patches of general application on diverse soft structures is very good and original. This idea is very powerful and the technology developed is also sound and impactful. They can be of interest in many fields. The work is done in a rigorous and thorough way and the results are complete and convincing. They are presented nicely and, together with a very clear description of the work, the paper is very readable and can reach a wide community. The supplementary material gives all the required details to understand and reproduce the work. The authors refer very much to the muscular hydrostat of the octopus, which is in fact very special because of its constant volume. With pneumatic actuation this assumption is not maintained and maybe some references to octopus arms (not tentacles) could be smoothed. The fact that the patches can help program a (even complex) movement is very interesting, but it still refers to movements that are somehow fixed by this programming. The octopus arm has some characteristic movements, like the reaching one, consisting of a bending wave from the base to the tip – is something like that reproducible with patches on a pneumatic arm? This is a genuine question that the authors may consider and maybe elaborate a bit in their discussion.

Reviewer #2 (Remarks to the Author):

This work presents a fabrication approach for creating soft actuators with controllable trajectories. Here, unidirectional fibers are wound into elastomeric layers which provides an anisotropic mechanical response which can be utilized for controlling the inflation characteristics of pneumatic actuators, the authors call this material a Stretchable Adhesive Uni-Directional prepreg (STAUD-prepreg). By attaching the STAUD-prepreg in different orientations on an inflatable balloon different actuation modes can be achieved which includes contraction, elongation, twisting and bending. Through more complex STAUD-prepreg layout combinations different deformation modes combine to provide more intricate trajectories, which is demonstrated in cylindrical actuators to mimic cephalopod tentacles. The experiments, modeling, and demonstrations are all very well done and data is presented clearly.

However, there are several previous studies which limit the conceptual or functional leap of the current study. As the authors point out in the introduction, there is a wide body of work which achieves diverse actuation geometries through the incorporation of a strain limiting layer or through an array of actuation chambers. Two notable examples which achieve an excellent variety of actuation trajectories include 'Robotic tentacles with three-dimensional mobility based on flexible elastomers' (ref 1 below) and 'Elastomeric Origami: Programmable Paper-Elastomer Composites as Pneumatic Actuators' (ref 2 below). These appear as references 14 and 22 in the current paper. There is also previous work on repositionable sleeves that allow for the actuation of soft bodies to be tuned (similar in concept although these are non-adhesive based) (ref 3 below). Fabric wrapped bladders have also been demonstrated to show that fabric anisotropy can provide different soft actuator deformations (ref 4 below).

In particular, ref 1 shows that air chambers can be incorporated at multiple locations along the actuator length to allow for selective pressurization and thus dynamic reconfigurability between multiple shapes. In the presented work, this capability is not present, and may be difficult to achieve as the STAUD-prepreg layers would have to be removed and then reattached in different configurations to achieve a variety of distinct shapes, while ref 1 has this inherently built in.

Further, ref 2 shows many of the same actuation modes presented in this work (contraction,

elongation, twisting and bending) through the incorporation of different geometries/orientations of paper. Although the current work uses uni-directional fibers as strain limiting layers instead of paper, both can be used to generate a variety of deformation modes and it is not clear based on the current manuscript text what advantage fibers would have for different applications over paper.

It should be made more clear how the current approach represents a break through over previous capabilities. Further, with the motivation of a cephalopod tentacle, can the authors predict what would be required in the presented framework to achieve an infinite degree of freedom actuator to mimic muscular hydrostats? Although it seems possible to remove and reattach STAUD-prepregs to achieve multiple configurations, the cephalopod can do this with a single tentacle without intervention, which is an advantage over current actuator systems.

Some other questions that should be addressed:

- Can the STAUD-prepreg layers be removed and reattached over multiple cycles? Are there limits?
- How reliable is the attachment of the STAUD-prepreg to the inflatable structure? Can this be used on the order of 100s or 1000s of cycles if large deformations are used?
- How do the properties of the inflatable soft body influence the actuation behavior? Are there limits for how rigid or soft the membrane must be relative to the STAUD? Some of the videos show a very large inflation of sections without STAUD-prepreg, could the properties of the soft body and STAUD-prepreg be matched in ways to reduce this effect?
- How does the shape of the soft body influence the final actuation shape (2D soft body vs cylinder vs sphere)?

1. Martinez, R. V. et al. *Adv. Mater.* 25, 205–212 (2013).
2. Martinez, R. V., Fish, C. R., Chen, X. & Whitesides, G. M. *Adv. Funct. Mater.* 22, 1376–1384 (2012).
3. K. C. Galloway, P. Polygerinos, C. J. Walsh, R. J. Wood, 2013 16th Int. Conf. Adv. Robot. ICAR 2013 2013, DOI 10.1109/ICAR.2013.6766586.
4. L. Cappello, K. C. Galloway, S. Sanan, D. A. Wagner, R. Granberry, S. Engelhardt, F. L. Haufe, J. D. Peisner, C. J. Walsh, *Soft Robot.* 2018, 5, 662.

Reviewer #3 (Remarks to the Author):

This paper reports a novel method to reconfigure soft body trajectories using unidirectionally stretchable composite laminae. The reviewer finds this article well written and results well presented. The reviewer recommends its publication with Nature Communications once the following issues are addressed:

1. "Mckibben" should be "McKibben" throughout the text.
2. "principle directions" should be "principal directions" on page 7 of the draft.
3. The technical issue is that the reconfiguration is static. Once the composite lamina is bonded to the inflatable soft body by a silicone adhesive, the soft body's deformation pattern is fixed. So seems like it is reconfigurable just for one time?
4. If one-time reconfiguration is intended, while the design is expected to be used multiple times, then its fatigue behavior, especially the interface between the lamina and the soft body, is not mentioned. It would be worthwhile to comment on this aspect.

We thank both the editor and reviewers for their time devoted to reading and providing insightful comments. Many good points were made, and we believe our revisions in response to these points will enhance the quality and impact of our manuscript. Below we provide a detailed point-by-point response to the reviewer comments.

Reviewer #1 (Remarks to the Author):

The paper presents an interesting laminated structure that can be applied to pneumatically-actuated soft robots in order to guide their deformation, thus the resulting movement of their actuation. The principle of using fibers to constrain and guide the deformation of pneumatically-actuated soft robots is not new per se, but the idea of building adhesive patches of general application on diverse soft structures is very good and original. This idea is very powerful and the technology developed is also sound and impactful. They can be of interest in many fields. The work is done in a rigorous and thorough way and the results are complete and convincing. They are presented nicely and, together with a very clear description of the work, the paper is very readable and can reach a wide community. The supplementary material gives all the required details to understand and reproduce the work.

The authors refer very much to the muscular hydrostat of the octopus, which is in fact very special because of its constant volume. With pneumatic actuation this assumption is not maintained and maybe some references to octopus arms (not tentacles) could be smoothed. The fact that the patches can help program a (even complex) movement is very interesting, but it still refers to movements that are somehow fixed by this programming. The octopus arm has some characteristic movements, like the reaching one, consisting of a bending wave from the base to the tip – is something like that reproducible with patches on a pneumatic arm? This is a genuine question that the authors may consider and maybe elaborate a bit in their discussion.

=> We agree with the reviewer regarding the nature of the constant volume maintained by the muscular hydrostat. The presented work is inspired by the structural morphology and muscle fiber configuration of the muscular hydrostat, as opposed to its movement that utilizes a lamination of unidirectional actively contracting layers without changing in volume. In contrast to hydrostat muscle fibers that actively contract, the embedded fibers in STAUD prepreg are entirely passive.

Thus, they require some external impetus to exert forces. We rely on the volumetric expansion of the host body, and the ensuing interaction between this expansion and the passive strain limiting fibers, to create contraction forces analogous to muscle fibers in the muscular hydrostat. We revised the abstract and introduction of the manuscript to clarify this point on page 2 line 8-10, page 3 line 23-page 4 line 2, and page 4 line 5-8.

To best mimic characteristic movements of the octopus's arm (*i.e.*, reaching, wave-type motions), re-distribution of local contraction forces is necessary. For example, muscular hydrostats can sequentially contract/relax local muscle fibers because individual fibers are independently controlled. Such locally-controlled motion can be realized when using a base pneumatic bladder consisting of multiple chambers with independent air lines or, alternatively, replacing passive polyester fibers with active fibers that change both the stiffness and direction of their fibers on-demand. We are leaving dynamic reconfigurability outside of the scope of the current paper. This paper is focused on a full analysis of fundamental mechanics and applications of STAUD-prepreg lamina/laminates, providing a solid foundation for future research. As a follow-up, we are currently working on a modified STAUD-prepreg that can modulate strain-limiting directions within a single patch by addressing embedded low-melting-point alloy fibers. We added current limitations and future directions to the discussion of the manuscript in page 9 line 15-19.

Reviewer #2 (Remarks to the Author):

This work presents a fabrication approach for creating soft actuators with controllable trajectories. Here, unidirectional fibers are wound into elastomeric layers which provides an anisotropic mechanical response which can be utilized for controlling the inflation characteristics of pneumatic actuators, the authors call this material a Stretchable Adhesive Uni-Directional prepreg (STAUD-prepreg). By attaching the STAUD-prepreg in different orientations on an inflatable balloon different actuation modes can be achieved which includes contraction, elongation, twisting and bending. Through more complex STAUD-prepreg layout combinations different deformation modes combine to provide more intricate trajectories, which is demonstrated in cylindrical actuators to mimic cephalopod tentacles. The experiments, modeling, and demonstrations are all very well done and data is presented clearly.

However, there are several previous studies which limit the conceptual or functional leap of the current study. As the authors point out in the introduction, there is a wide body of work which achieves diverse actuation geometries through the incorporation of a strain limiting layer or through an array of actuation chambers. Two notable examples which achieve an excellent variety of actuation trajectories include ‘Robotic tentacles with three-dimensional mobility based on flexible elastomers’ (ref 1 below) and ‘Elastomeric Origami: Programmable Paper-Elastomer Composites as Pneumatic Actuators’ (ref 2 below). These appear as references 14 and 22 in the current paper. There is also previous work on repositionable sleeves that allow for the actuation of soft bodies to be tuned (similar in concept although these are non-adhesive based) (ref 3 below). Fabric wrapped bladders have also been demonstrated to show that fabric anisotropy can provide different soft actuator deformations (ref 4 below).

In particular, ref 1 shows that air chambers can be incorporated at multiple locations along the actuator length to allow for selective pressurization and thus dynamic reconfigurability between multiple shapes. In the presented work, this capability is not present, and may be difficult to achieve as the STAUD-prepreg layers would have to be removed and then reattached in different configurations to achieve a variety of distinct shapes, while ref 1 has this inherently built in. Further, ref 2 shows many of the same actuation modes presented in this work (contraction, elongation, twisting and bending) through the incorporation of different geometries/orientations of paper. Although the current work uses uni-directional fibers as strain limiting layers instead of

paper, both can be used to generate a variety of deformation modes and it is not clear based on the current manuscript text what advantage fibers would have for different applications over paper.

It should be made more clear how the current approach represents a break through over previous capabilities. Further, with the motivation of a cephalopod tentacle, can the authors predict what would be required in the presented framework to achieve an infinite degree of freedom actuator to mimic muscular hydrostats? Although it seems possible to remove and reattach STAUD-prepregs to achieve multiple configurations, the cephalopod can do this with a single tentacle without intervention, which is an advantage over current actuator systems.

=> We thank reviewer for pointing a number of important points out. We believe that the presented work addresses three major obstacles in state-of-art soft robotics: 1) lack of rapid, scalable, and high-throughput manufacturing processes, 2) permanent shape morphing determined at fabrication, 3) difficulty in analytically predicting shape change of morphing of soft bodies. Addressing the first point: STAUD-prepreg is fabricated via a fiber winding process that is already well-established in industry for mass-production of industrial-grade composites, but up to now has not been adapted to the soft robotics space for soft materials. As a direct consequence of this novel manufacturing technique, we can realize a highly customized two-dimensional (2D) material that can be adhered to almost any soft actuator—be it 2D or 3D—to control its motion.

For the second point, as the reviewer indicated, re-adjusting the stacking sequence, localization, and areal coverage of STAUD-prepreg reconfigures the inflation trajectory of the soft body they are attached to—even if this soft body has only one internal bladder. In addition, by simply removing and re-attaching the STAUD-prepreg varying the listed parameters above, a totally new type of motion can be achieved by the same exact soft body. We showcase how easy it is to remove and re-attach STAUD prepreg to elicit different deformation modes in supplementary video 5. With such fine surficial strain limiting capability, achievable deformation trajectories are nearly infinite in number, and reach a level of tuneability that is simply not possible with existing strain limiting approaches.

For the third point, the fact that STAUD-prepreg are laminates composed of individual unidirectional lamina allows us to utilize a variant of classical laminate theory to analytically predict the shape morphing. This analytical method is relatively easy to access and serves as a powerful tool when designing soft robots. We further emphasized and clarified these

breakthroughs (esp. manufacturing benefits) in the introduction of the manuscript, page 4 line 8-11.

To compare with the references that the reviewer provided: Ref 1 is a valuable foundation upon which the current work builds (and cites accordingly). Ref 1 showcased how very complex and previously unattainable deformation could arise from creating bulk actuator structures from different stiffnesses of elastomer. One challenge associated with the approach is that it would be difficult to take the reported process to a mass manufacturing scale; in particular the authors report 3 hours to realize a single actuator and this time increases as the number of pneumatic chambers increases. Another contribution of Ref 1 is the notion of reconfigurable trajectories accomplished via a number of independently addressed airlines and chambers. Building upon Ref 1, our system reaches the next step of generalizability since it uses only a single inflatable chamber and a single airline to render sophisticated movement. Furthermore, STUAD-prepreg can be adhered to a system with multiple pneumatic chambers, like those reported in Ref 1, and achieve reconfigurable trajectories more predictably (due to the analytically predictable nature of the unidirectional strain limiting laminae).

Ref 2, like Ref 1, introduced fundamental concepts to the soft actuator design space that were a source of inspiration for the presented manuscript. Inserting folded paper of tailored dimensions and shapes into elastomeric structures, Ref 2 achieved actuators with a host of useful and sophisticated movements. One way our work extends the results of Ref 2 is via a scalable manufacturing process, since each paper in Ref 2 needs to be laser-cut or folded, and subsequently embedded into elastomeric structures. STAUD-prepreg, in contrast, is fabricated en-masse and adhered to a soft body's surface with minimal effort. Another point our work extends from Ref 2 is related to mechanical uniformity. The inserted papers in Ref 2 are 2D elements (sheets of width > 1.5 mm) and occupy a substantial foot-print that compromises the uniformity of the mechanical properties of the actuator (*i.e.*, paper covers large area that prevents many parts of the matrix from stretching). The work presented here utilizes uniformly embedded 1D elements (fibers with diameter < 0.1 mm) with a fiber spacing > 1 mm. This design facilitates a high degree of matrix stretching where it is desired in a structure, yet precise control over the strain limiting direction.

Ref 3—like Ref 1 and Ref 2—presents fundamental, contributing work in that it uses adjustable sleeves to constrain the bending of specific locations along the length of an actuator. However, the sleeve is designed to completely lock out specific sections, rather than control the

trajectory of specific locations within those sections. This approach is conceptually different from ours, although both impact the overall trajectory of the actuator. We note that the adjustable sleeves might be challenging to implement on planar bladders, and that the STAUD-prepreg was designed to be easy to be attached to any soft actuator regardless of actuator morphology.

We feel that Ref 4 can serve as a good example that uses the directionality of stiffness and cite it in the present manuscript as [19]; we thank the reviewer for drawing this work to our attention. Ref 4 utilized the difference in the stretchability of woven fabrics and knit fabrics to create bending motions. Our work builds upon this previous study by showcasing a multitude of complex motions via simple changes in fiber orientation and localization. Moreover, STAUD-prepreg exhibits a 1000-fold difference in stiffness between the fiber direction and the normal of this direction, which maximizes its strain-limiting effect.

To achieve an infinite degree of freedom actuator and mimic the diversity of motion witnessed in muscular hydrostats, re-distribution of local contraction forces is necessary. For example, muscular hydrostats can sequentially contract/relax local muscle fibers because individual fibers are independently controlled. Such locally-controlled motion can be realized when using a base pneumatic bladder consisting of multiple chambers with independent air lines or, alternatively, replacing passive polyester fibers with active fibers that change both the stiffness and direction of their fibers on-demand. We are leaving such a next level of study outside of the scope of the current paper. This paper is focused on a full analysis of fundamental mechanics and applications of STAUD-prepreg lamina/laminates, providing a solid foundation for future research. As a follow-up, we are currently working on a modified STAUD-prepreg that can modulate strain-limiting directions within a single patch by addressing embedded low-melting-point alloy fibers. We added current limitations and future directions to the discussion of the manuscript in page 9 line 15-19.

Some other questions that should be addressed:

- Can the STAUD-prepreg layers be removed and reattached over multiple cycles? Are there limits?
=> Yes, STAUD-prepregs can be removed and reattached over multiple cycles. The limit is determined by the preparation of the surface (*i.e.*, cleanliness, roughness) to where they are attached. A recent study (Liu et al., Biomed. Phy. & Eng. Exp., 2017) examined the self-adhesive

layer we used and reported relatively consistent adhesive force up to 20 cycles. We added this point to the manuscript in page 10 lines 20-22, and referenced the relevant study as [39].

- How reliable is the attachment of the STAUD-prepreg to the inflatable structure? Can this be used on the order of 100s or 1000s of cycles if large deformations are used?

=> The adhesion of STAUD-prepreg onto silicone substrate is reliable up to the certain degree of deformation, as shown in Fig. 3b(i). The figure shows the detachment of the prepreg at a low surface coverage of 16 %. This detachment demonstrates that surface areal coverage of the prepreg on the soft body can be increased to enhance adhesion, thereby prolonging attachment. Within the reliable attachment region (dependent on a combination of areal coverage and deformation), we have not observed any prepreg detachment from a host body throughout our tests that applied inflation/deflation motion on the order of 100s cycles. Such persistent attachment is presumably due to the characteristics of gel of the adhesive layer in that it can stretch to large strains without tear or damage. We added this detail to the manuscript page 10 line 22-page 11 line 2.

- How do the properties of the inflatable soft body influence the actuation behavior? Are there limits for how rigid or soft the membrane must be relative to the STAUD? Some of the videos show a very large inflation of sections without STAUD-prepreg, could the properties of the soft body and STAUD-prepreg be matched in ways to reduce this effect?

=> The reviewer is correct. The softness of the base bladder determines how much deformation an adhered STAUD-prepreg can undergo. At a given deformation, the higher the stiffness of the base bladder, the higher an adhesive force is required for the STAUD-prepreg to remain adhered and effectively control the deformation. For example, we tried using a base bladder consisting of stiffer Dragon Skin 10 (shore hardness 10A) instead of Eco-flex (shore hardness 00-30), and found that STAUD-prepreg detached more easily from Dragon Skin 10. We have not performed a full analysis of such limitations arising from disparate stiffnesses of STAUD-prepreg and the base material, because the presented paper is largely focused on the mechanical influence of fiber orientations. We believe the fundamental analysis herein is a stepping stone to further research regarding different types of matrices and reinforcing fibers to even further tune mechanical properties (*i.e.*, stiffness) of STAUD-prepreg constituents. We added this point to the material section of the manuscript page 10 line 4-6.

We think the large inflation of bare elastomer matrix noted in the video can be efficiently controlled using another STAUD-prepreg to prevent excessive radial expansion, rather than

matching the material stiffnesses. After all, the inflation is a source of the actuation force. For example, additional STAUD-prepreg can be adhered onto the section at a 90° fiber orientation, then both top and bottom surfaces will be guided to a simple bending without unnecessary large inflations.

- How does the shape of the soft body influence the final actuation shape (2D soft body vs cylinder vs sphere)?

=> We surmise that the initial shape of soft bodies (2D vs 3D cylinder, sphere) influences more the adherence of the STAUD-prepreg than it does the uniqueness of a final shape. For example, the presented study is performed with 2D bodies and 3D cylinders; we found that STAUD-prepreg are easier to adhere to 2D bodies than 3D ones without void entrapments, because both the base and prepreg are in planar configuration. Entrapped voids degrade the adhesion quality and result in premature detachment of the STAUD-prepreg, so the body cannot reach its final intended shape. Otherwise, the final shape was more determined by the fiber orientations of the STAUD-prepreg and their local arrangement on the surface of a bladder rather than the initial shapes of the bladders themselves. We added this detail to the manuscript page 5 line 13-16.

1. Martinez, R. V. et al. *Adv. Mater.* 25, 205–212 (2013).
2. Martinez, R. V., Fish, C. R., Chen, X. & Whitesides, G. M. *Adv. Funct. Mater.* 22, 1376–1384 (2012).
3. K. C. Galloway, P. Polygerinos, C. J. Walsh, R. J. Wood, 2013 16th Int. Conf. Adv. Robot. ICAR 2013 2013, DOI 10.1109/ICAR.2013.6766586.
4. L. Cappello, K. C. Galloway, S. Sanan, D. A. Wagner, R. Granberry, S. Engelhardt, F. L. Haufe, J. D. Peisner, C. J. Walsh, *Soft Robot.* 2018, 5, 662.

Reviewer #3 (Remarks to the Author):

This paper reports a novel method to reconfigure soft body trajectories using unidirectionally stretchable composite laminae. The reviewer finds this article well written and results well presented. The reviewer recommends its publication with Nature Communications once the following issues are addressed:

1. "Mckibben" should be "McKibben" throughout the text.

=> We thank the reviewer for finding this out. The terminology is rectified.

2. "principle directions" should be "principal directions" on page 7 of the draft.

=> The terminology is rectified.

3. The technical issue is that the reconfiguration is static. Once the composite lamina is bonded to the inflatable soft body by a silicone adhesive, the soft body's deformation pattern is fixed. So seems like it is reconfigurable just for one time?

=> The STAUD-prepreg can be detached and attached multiple times; it is not permanently bonded to the surface. Rather, it is a temporary (but sufficiently strong) bond using a self-adhesive back layer. We found that this feature of the STAUD-prepreg was not fully elaborated upon in the introduction section, and made appropriate revisions in the manuscript page 4 lines 13-14 and 15-17.

4. If one-time reconfiguration is intended, while the design is expected to be used multiple times, then its fatigue behavior, especially the interface between the lamina and the soft body, is not mentioned. It would be worthwhile to comment on this aspect.

=> We agree with the reviewer. Although STAUD-prepreg is a re-adjustable patch rather than a permanent fixture, its interfacial bonding to soft substrates are critical. The interfacial bonding over multiple cycles of detachment/attachment is determined by the preparation of the surface (*i.e.*, cleanliness, roughness) and to where they are attached. A recent study (Liu et al., Biomed. Phy. & Eng. Exp., 2017) examined such characteristics of the self-adhesive layer we used and reported relatively consistent adhesive force up to 20 cycles. We added this point to the manuscript in page 10 line 20-22, and referenced the relevant study as [39].

REVIEWERS' COMMENTS:

Reviewer #1 (Remarks to the Author):

The authors submitted a revised version that complies with the reviewers' comments, together with the explanations given.

Concerning the muscular hydrostat, I still think that the bioinspiration is not central and it could be smoothed even more. However, the work itself is interesting and deserves publication.

Reviewer #2 (Remarks to the Author):

The authors have addressed the points from the reviewers. The paper is acceptable for publication.

Reviewer #3 (Remarks to the Author):

The authors have addressed the issues raised by the reviewers, and thus the paper is recommended to be published with Nature Communications.